# Reconfigurable Microwave Multi-Beamforming Based on Optical Switching and Distributing Network

**Yue Lin, Di Jiang, Yuan Chen, Xiang Li** and **Qi Qiu** *

The School of Optoelectronic Science and Engineering, University of Electronic Science and Technology of China, Chengdu 610054, China; linyue0204@163.com (Y.L.); jiangdi0313@gmail.com (D.J.); cy136798606@163.com (Y.C.); lixiang3661@163.com (X.L.)
* Correspondence: qqiu@uestc.edu.cn

**Abstract:** Optical beamforming in microwave photonics is promising for supporting broadband wireless communications. However, the current optical beamforming lacks freedom because of the fixed connection between radio frequency (RF) signal and antenna elements (AEs). This manuscript tackles this challenge by proposing a dynamical optical beamforming architecture that reconfigures the antenna subarray for signal transmission depending on the number of signals to be transmitted. The proposed architecture employs an optical switching and distributing network (SDN) to realize a flexible connection between signals and AEs. An instance of the proposed architecture in photonic integrated circuits, which enables three working modes and transmits four RF signals through sixteen AEs, was presented and numerically simulated. The optical field distribution and beam pattern plots illustrated the operational principle and validated the feasibility of the proposed SDN architecture. Furthermore, the impact of the introduced architecture on the signal amplitude–phase consistency and the comparison of the proposed dynamic architecture and conventional fixe architectures are analyzed and discussed. The results indicate that the proposed architecture exhibits variable beamforming gain with lower hardware complexity.

**Keywords:** microwave photonics; beamforming; reconfigurable



## 1. Introduction

Higher radio frequency (RF) bands between 24.25 GHz and 86 GHz are opened in fifth-generation (5G) communication to satisfy the increasing demands of the throughput in wireless communication [1]. The RF bands are further increased to 100 GHz in sixth-generation (6G) communication. However, the higher RF bands lead to a smaller signal beam coverage. To improve the signal coverage efficiency, the multiple antenna technology was introduced into wireless communication at the earliest in Release 7 by 3GPP [2]. The multiple antenna technology not only offers the beamforming gain in the conventional phased array antenna but also needs to offer the spatial multiplexing gain and diversity gain, which improve the system capacity and spectral efficiency.

In the conventional phased array antenna, the RF signal from the transmitter is fed to numerous individual antenna elements (AEs) with the proper phase relationship manipulated by the electronic phase shifter, so that the RF signal from the separate AEs combine to form beams and focus power radiated in desired directions [3,4]. Nevertheless, when the RF signal frequency is expanded into the millimeter-wave band, the conventional electronic phase shifter will lead to the beam squint effect, which limits the system operation for broadband [5]. Providing the true-time delay (TTD) to each AE can prevent beam direction changes with signal frequency changes and eliminate the beam squint effect. Electronic TTD, due to the high loss property of coaxial cable and dielectric mediums, is energy inefficient. Conversely, optical TTD (OTTD), with its benefits of low loss, broadband, and immunity to electromagnetic interference, is more appropriate for future high-frequency wireless communication [5–7].

In general, the research on OTTD can be categorized into two types according to whether the delay value is independent of the optical carrier wavelength [8–16]. In the first type, the delay values are independent of the optical carrier wavelength. This type possesses a large delay bandwidth product. The OTTD line, which uses the optical switch to redirect the optical path of various discrete lengths, is a representative architecture of this type [8–10]. The other type of generating delay values is dependent on the optical carrier wavelength, such as the micro-ring resonator [11,12], the loop waveguide based on the array waveguide grating [13], and the multiple wavelength group delay based on the dispersive material [14,15] or optical grating [16]. The second type usually utilizes wavelength division multiplexing (WDM) technology, which implies a finite delay bandwidth and limits the operating bandwidth.

However, the above architectures can only transmit the RF signals to fixed-connected AEs. Therefore, the current optical beamforming system lacks the degree of freedom and fails to realize emergent technologies, such as multi-function radar [17] and antenna selection [18]. Two typical fixed-connected architectures are illustrated in Figure 1, referred to as fully and partially connected, respectively [19]. In the fully connected architecture, each RF signal has its own OTTD array and is connected to all the AEs in the antenna array. In contrast, each RF signal in the partially connected architecture is connected to a subarray of AEs. The fully connected architecture has the highest beamforming gain and hardware complexity. Compared with the fully connected architecture, the partially connected architecture cannot achieve the full beamforming gain but reduces hardware complexity.

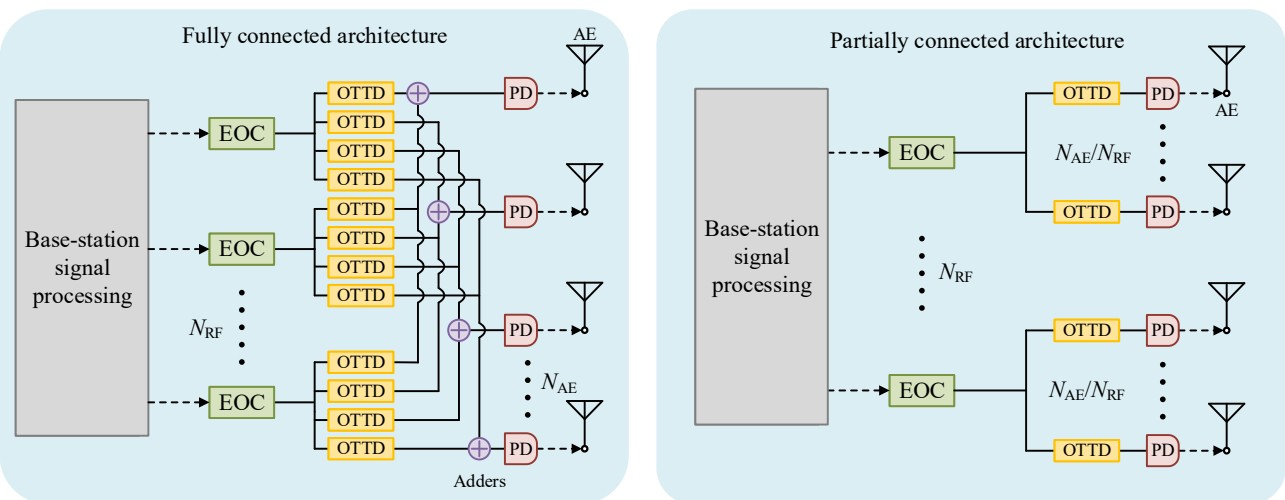

**Figure 1.** Illustration of two typical fixed-connected architectures. EOC: electronic–optical conversion; OTTD: optical true-time delay; PD: photodetector; AE: antenna element; $N_{RF}$: the number of radio frequency (RF) signals to transmit; $N_{AE}$: the number of antenna elements.

This manuscript proposed a dynamical optical beamforming architecture to add a new degree of freedom to the optical beamforming system, breaking the limitations in the fixed-connected architecture and adapting them to meet the hardware requirement of modern wireless communication. The reconfigurable connections between RF signals and the antenna subarray are realized by integrating the optical switching and distributing network (SDN). The proposed dynamical optical beamforming architecture is theoretically analyzed in Section 2 and verified in Section 3. The impact of the proposed dynamical architecture on the signal amplitude–phase consistency and the comparison of the proposed dynamic architecture and conventional fixed architectures are analyzed and discussed in Section 4. The conclusion is given in Section 5.

## 2. System Architecture

Figure 2 shows the reconfigurable optical multi-beamforming transmitter based on the SDN. The system comprises electronic–optical conversion, SDN, OTTD array, optoelectronic conversion, and RF signal transmitter.

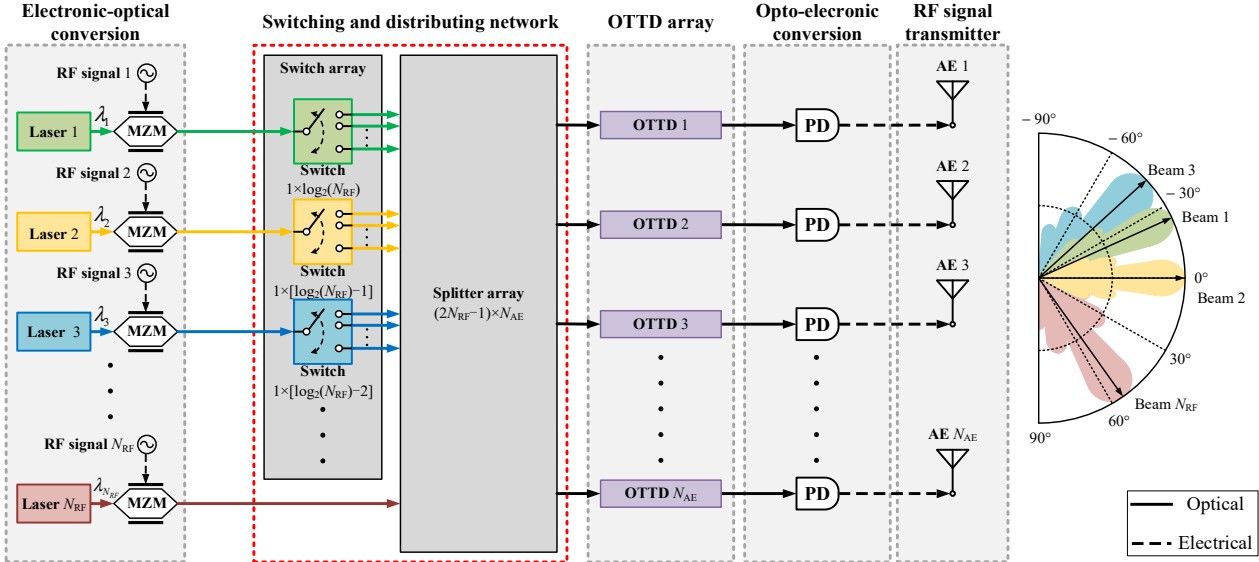

**Figure 2.** Schematic of the optical reconfigurable multi-beamforming architecture based on switching and distributing network. $\lambda_n$: the wavelength of optical carrier $n$; MZM: Mach–Zehnder modulator.

The most critical part of the designed system is the SDN, which consists of the optical switch array and the splitter array. By controlling the optical switch array, the optical signals can perform different behaviors in the splitter array, reconfiguring the connections between the RF signals and the antenna subarrays. Consequently, by controlling the working mode of the SDN depending on the number of RF signals inputted into the system, the input RF signals can be transmitted to suitable antenna subarrays. The functional and operational principles of each part will be detailed below according to the flow of signals.

### 2.1. Electronic–Optical Conversion

The proposed optical beamformer starts by modulating the RF signal onto optical carriers. The electronic–optical conversion contained numerous groups of lasers and modulators. The number of the groups is $N_{RF}$, equaling the maximum number of simultaneous input RF signals supported by the transmitter. The different-group lasers are set to be different wavelengths with identical intervals. The different wavelengths of each optical carrier are selected to prevent crosstalk resulting from optical component imperfections in the subsequent optical path, which ultimately affects signal transmission and processing.

A laser generates an optical carrier $E_0(t)$ with a given wavelength ($\lambda_n$). Then, using a Mach–Zehnder modulator (MZM) biased in the quadrature point, an RF signal $S(t)$ with the angular frequency of $\omega_{RF}$ is converted into the optical domain. Hence, the optical signal $E(t)$ can be obtained at the output of MZM, which can be expressed as [20]:

$$\begin{cases} E_o(t) = A_o \exp\left(-j\frac{2\pi c}{\lambda_n}t\right) \\ S(t) = A(t) \exp[j\omega_{RF}(t)t] \\ E(t) = E_o(t) \cos\left[\frac{\pi}{4} + \frac{\pi}{2V_\pi}S(t)\right] \end{cases} \tag{1}$$

where $A_0$ represents the amplitude of the optical carrier, $A(t)$ and $\omega_{RF}(t)$ describe the instant amplitude and frequency of the signal, respectively, $j$ is the imaginary unit, and $V\pi$ is the driver voltage of the modulator. $N$ optical signals with various optical wavelengths

are sent into the subsequent optical path and have the signal routing and processing, $N \in [1, N_{RF}]$.

### 2.2. Optical Switch Array

After modulation, the modulated optical signals are fed into the optical switch array. A single optical switch array is illustrated in Figure 3, constituted by $1 \times 2$ optical switch elements and arranged in the binary trees.

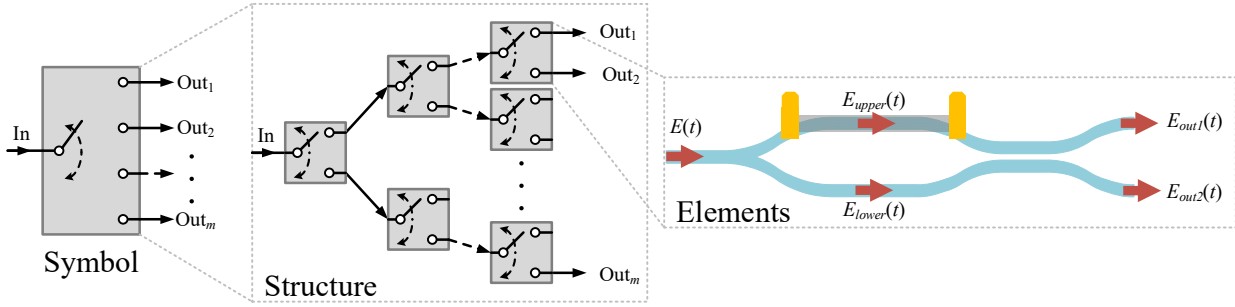

**Figure 3.** The schematic diagram of the optical switch array.

The behavior of optical signal $E(t)$ sent into a single $1 \times 2$ optical switch element can be described as follows: firstly, the signal passes the y-branch and splits into two arms. The optical field in the upper arm and lower arm can be expressed as follows:

$$E_{upper}(t) = E_{lower}(t) = \frac{\sqrt{2}}{2}E(t) \tag{2}$$

Then, a phase difference $\Delta\varphi$ between two arms is achieved by carrier injection or thermo-optic effects. Finally, the two signals are sent into a 3 dB directional coupler (DC), and the optical field in the upper and the lower output can be formulated as follows:

$$\begin{cases} E_{out1} = \frac{1}{2}E(t)\exp[-j(\Delta\varphi)] + \frac{1}{2}E(t)\exp[-j(\frac{\pi}{2})] \\ E_{out2} = \frac{1}{2}E(t)\exp[-j(\Delta\varphi + \frac{\pi}{2})] + \frac{1}{2}E(t) \end{cases} \tag{3}$$

where the phase shift $\pi/2$ is caused by coupling [21]. Thus, the optical intensity can be calculated based on Equation (3) and expressed as follows:

$$\begin{cases} I_{out1} = E_{out1}E_{out1}^\dagger = \frac{1}{2}E^2(t)(1 + \sin\Delta\varphi) \\ I_{out2} = E_{out2}E_{out2}^\dagger = \frac{1}{2}E^2(t)(1 - \sin\Delta\varphi) \end{cases} \tag{4}$$

where the superscript '†' represents the operation of complex conjugation. By changing the phase difference $\Delta\varphi$ between two arms as $+\pi/2$ or $-\pi/2$, the optical energy of two outputs can be switched. Based on that, the optical signal routing can be accomplished using the optical switch array. It should be noted that, for an optical switch array with a total of $M$ output ports, the quantity of $1 \times 2$ optical switch elements included is $M - 1$.

### 2.3. Splitter Array

Figure 4 shows the splitter array, which is constituted by multiple stages of $1 \times 2$ splitters and $2 \times 2$ couplers arranged as binary trees. The proposed splitter array is realized using the $2 \times 2$ couplers to replace the $1 \times 2$ splitters from stage 2 to stage $\log_2(N_{RF}) + 1$ in a conventional splitter with multiple outputs. The entire splitter array can be divided into two sections according to the function of each section. The first section is the access section, which spans from stage 1 to stage $\log_2(N_{RF}) + 1$ and is designated for injecting the optical signals. The second section is the split section, which contains stage $\log_2(N_{RF}) + 2$ to $\log_2(N_{AE})$ and is only used to split the optical signals. $N_{AE}$ is the number of antenna elements.

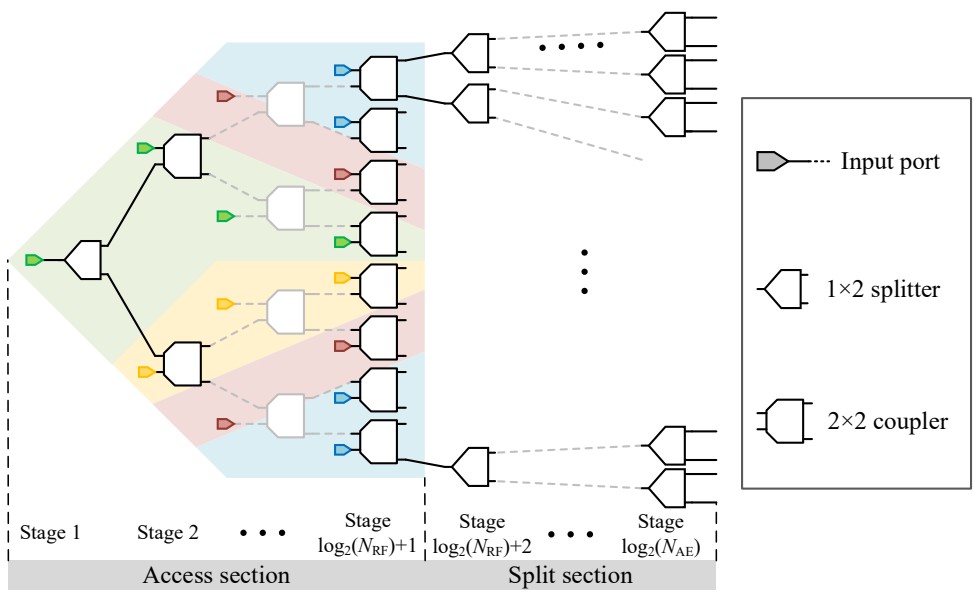

**Figure 4.** The schematic diagram of the splitter array.

Some details can be observed in Figure 4 and should be noted. Firstly, the various block colors indicate the use of the splitter or coupler to inject signals with different priorities. The input ports in the same color are used to inject the signal with the same priority. The signal priority indicates the maximum number of AEs enabled by the signal. A higher signal priority denotes that the signal can enable more AEs. Secondly, implementing a $2 \times 2$ coupler in the system will result in a phase shift among the output ports. Thus, the phase shifters are employed at the junction between two functional sections to compensate for phase inconsistency. Lastly, the splitter and coupler have a high insertion loss. Therefore, the power compensation should also be deployed at the junction between the two functional sections.

The splitting behaviors of the optical signal in the splitter array and the connection between optical signals and AEs are changed with the input stage order that the optical signal injects the splitter array. A specific optical path is created when the optical signal injects the splitter array. The number of splitters and couplers included in this optical path is determined. Therefore, by manipulating the aforementioned optical switch array, the optical signal can inject the splitter array at different stage orders and reconfigure the optical path, including different splitter and coupler numbers. Finally, the reconfigurable connection between signals and AEs is realized. Based on the above principle, the output ports of the optical switch array should be connected to the input ports of the splitter array at different stage orders according to the signal priority. The relationship and rules are listed in Table 1. In particular, Figure 5 illustrates the connection between the optical switch array and splitter array in the proposed architecture with $N_{RF} \geq 8$.

**Table 1.** The connection rules between the optical switch array and splitter array.

| Priority Rank * | Series Number of Signals | Number of Signals of the Same Priority | Stage Order Connected | Number of Switch Array Outputs |
|---|---|---|---|---|
| 0 | 1 | 1 | $1, 2, \ldots, \log_2(N_{RF}) + 1$ | $\log_2(N_{RF}) + 1$ |
| 1 | 2 | 1 | $2, \ldots, \log_2(N_{RF}) + 1$ | $\log_2(N_{RF})$ |
| 2 | 3, 4 | 2 | $3, \ldots, \log_2(N_{RF}) + 1$ | $\log_2(N_{RF}) - 1$ |
| $k$ | $2^k + 1, \ldots, 2^{k+1}$ | $2^k$ | $k + 1, \ldots, \log_2(N_{RF}) + 1$ | $\log_2(N_{RF}) - k + 1$ |
| $\log_2(N_{RF}) - 1$ | $N_{RF}/2, \ldots, N_{RF}$ | $N_{RF}/2$ | $\log_2(N_{RF}) + 1$ | 1 |

* The value of priority rank is smaller, indicating that the current signal has a higher priority and can enable more AEs.

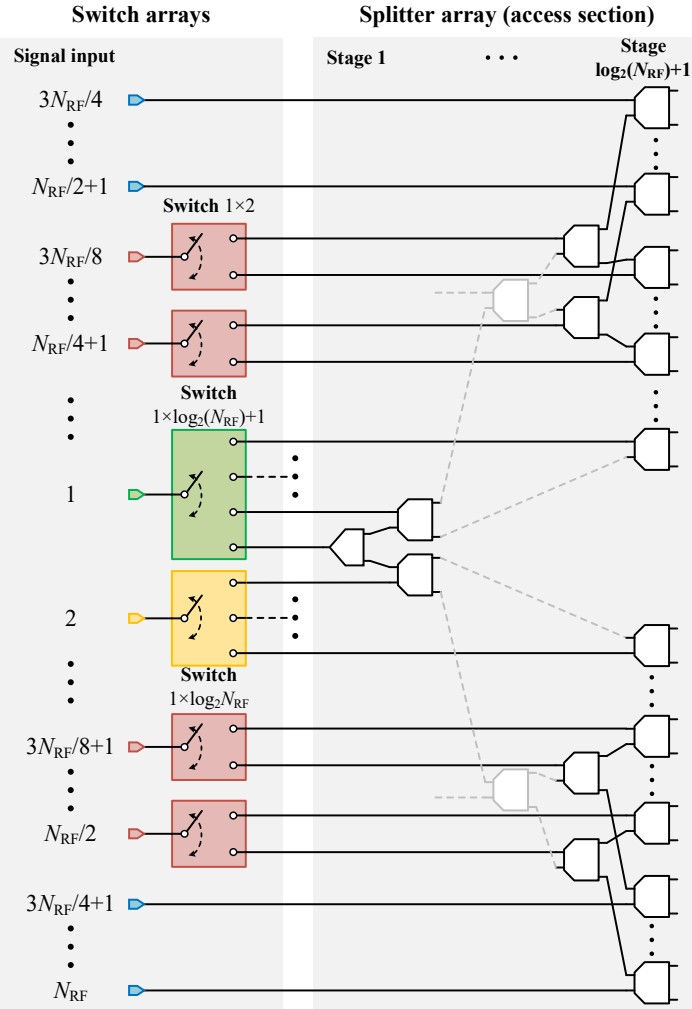

**Figure 5.** The illustration of the connection between the optical switch array and splitter array.

### 2.4. OTTD Array

An OTTD array is placed after the splitter array. OTTD devices available here include optical true-time delay lines [22], optical resonance rings [23,24], and other structures. Thanks to the advantage of optical true-time delay lines, such as large bandwidth and large delay range, they are widely used in many situations [25]. In our proposal, the true-time delay lines are also chosen to provide the true-time delay value in optical signals at each branch. The delay value of branch $i$ can be denoted as $\Delta\tau_i$.

### 2.5. Optoelectronic Conversion

The optoelectronic conversion part consists of $N_{AE}$ photodetectors. Each photodetector is used to down convert the optical delayed signal of each branch back into the electronic domain. The electrical signal of branch $i$ can be expressed as follows [20]:

$$\begin{aligned} I_i(t) &= \Re|E_i(t)|^2 \\ &\propto \frac{\Re}{2}\left[1 - \sin\left(\frac{2\pi}{V_\pi}S(t - \Delta\tau_i)\right)\right] \end{aligned} \quad (5)$$

where $\Re$ is the responsivity of the photodetector.

### 2.6. RF Signal Transmitter and Beamforming

The optical true-time delayed electrical signal will be radioed into the free space at the AEs. Since the switch array and splitter array enable reconfigurable signal switching

and distribution, the RF signal can only be sent through the AEs for which a connection exists under the current configurations. The far-field beam pattern of RF signal $n$ can be calculated by the following equation:

$$F_n(\theta) = \sum_{(a_i, \Delta\tau_i) \in \mathbf{S}_n} a_i \exp\left[-j2\pi f\left(\frac{d}{c}\sin\theta - \Delta\tau_i\right)\right], n \in [1, N] \tag{6}$$

where $\mathbf{S}_n$ represents the set of RF signal $n$, in which the element is expressed as a parameter pair $(a_i, \Delta\tau_i)$, and $a_i$ represents the excitation power of AE $i$.

## 3. Simulation

A series of numerical simulations were performed to demonstrate the principle of operation and validate the feasibility of the proposed architecture.

The antenna array employed in the transmitter is a uniformly distributed linear array containing 16 AEs, which are spaced by half the RF signal wavelength. The frequency of the RF signal is set at 28 GHz. Thus, the antenna element spacing is 4 mm. In addition, the maximum number of RF signal inputs is set to $N_{RF} = 4$. Therefore, the optical SDN, which supports 4 RF signals beamforming and transmitting by 16 AEs, is shown in Figure 6.

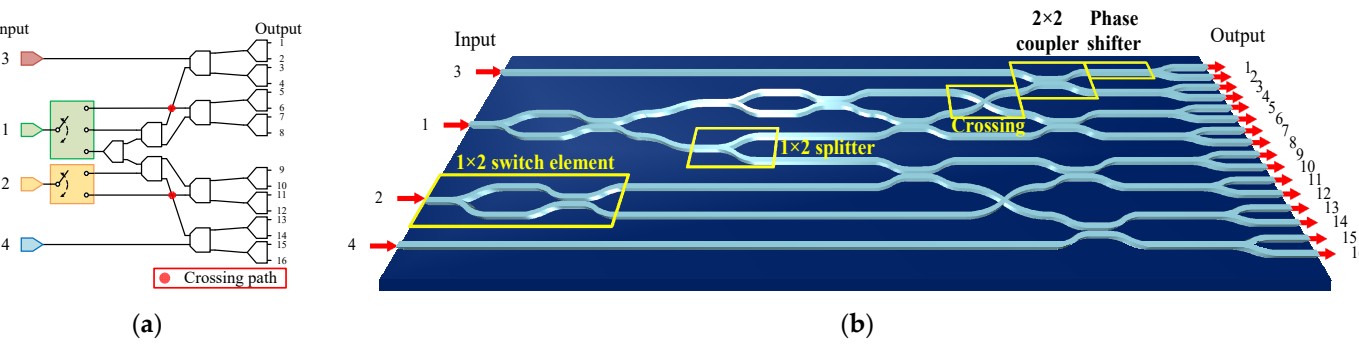

**Figure 6.** (**a**) Block diagram of $4 \times 16$ switching and distributing network; (**b**) the photonics integrated circuit layout.

The proposed architecture in the photonic integrated circuit is simulated to verify its feasibility. Waveguide parameters adhere to [26] and are 0.22 μm in height and 0.5 μm in width on the SOI platform. It can be seen in Figure 6b that the $4 \times 16$ optical SDN consists of three $1 \times 2$ optical switch elements, nine $1 \times 2$ splitters, six $2 \times 2$ directional couplers, and two waveguide crossings. The structure design of the $1 \times 2$ splitter is from [27], the $2 \times 2$ directional coupler is from [28], and the waveguide crossing is from [29]. The individual simulation of optical components employed in SDN was performed, and the S-parameter response results are shown in Figure 7. For 3-port components, port 1 is the input port, port 2 is the upper output, and port 3 is the lower output. For 4-port components, port 1 is the input port, port 2 is the upper output at the opposite side to port 1, port 3 is the lower output at the opposite side to port 1, and port 4 is the lower output at the same side to port 1. The parameter details of each optical component are listed in Table 2.

**Table 2.** The parameter setup of optical components.

| Components | Insertion Loss | Extra Loss | Crosstalk |
|---|---|---|---|
| $1 \times 2$ switch | 0.80 | - | 40 |
| $1 \times 2$ splitter [27] | 3.01 | 0.30 | - |
| $2 \times 2$ coupler [28] | 3.01 | 0.50 | - |
| Waveguide crossing [29] | 0.03 | - | 27 |

All the values are in dB.

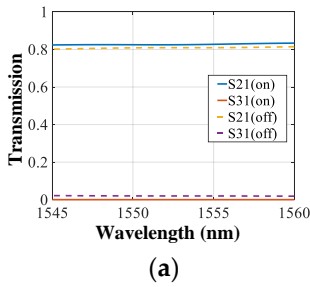
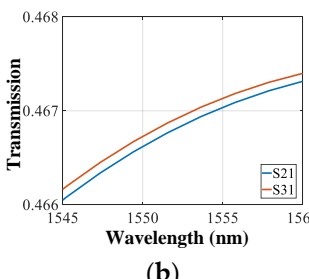
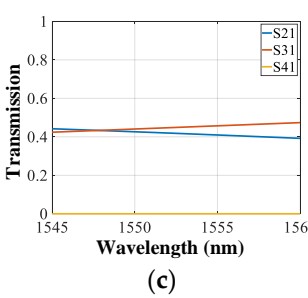
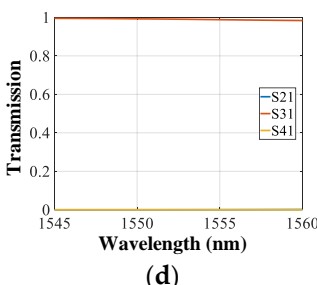

|                                  |                                  |
| :------------------------------: | :------------------------------: |
| **(a)**                          | **(b)**                          |

**Figure 7.** The S-parameter response of individual components; (**a**) the $1 \times 2$ optical switch elements; (**b**) the $1 \times 2$ splitters; (**c**) the $2 \times 2$ directional coupler; (**d**) the waveguide crossing.

Controlling three $1 \times 2$ optical switch elements enables the multi-beamforming system to operate in three modes: single-beam, dual-beam, and quadra-beam. The four RF signals are modulated on four optical carriers with the optical wavelengths 1550, 1551, 1552, and 1553 nm, respectively. Therefore, the optical power distribution and the corresponding beam pattern were plotted for each working mode to evaluate the impacts of implementing the optical switching and distributing network to the beamforming system.

Given that the main purpose of the simulations is to demonstrate the technical feasibility and evaluate the effects of optical SDN, the OTTD devices employed in the proposed architecture are the same as the OTTD devices employed in the architecture in Figure 1, which can be seen as an ideal component. They can accurately provide the true-time delay value to the AEs and steer the beam toward eleven directions as $-45.00°$, $-34.45°$, $-25.10°$, $-16.45°$, $-8.15°$, $0°$, $8.15°$, $16.45°$, $25.10°$, $34.45°$, and $45.00°$. The reconfigurable multi-beamforming system forms and steers the beams with different numbers in three working modes. In the simulation, the situations were set as follows: In the single-beam situation, only one signal beam is formed with a direction of $-34.45°$. In the dual-beam working mode, two beams are formed. The directions of the two signal beams are $-34.45°$ and $16.45°$, respectively. In the quadra-beam mode, four signal beams are formed with directions of $-34.45°$, $16.45°$, $-8.15°$, and $25.10°$, respectively. The related true-time delay values in three working modes are listed in Table 3.

**Table 3.** The related true-time delay values provided by OTTD devices.

| Outputs | | 1 | 2 | 3 | 4 | 5 | 6 | 7 | 8 | 9 | 10 | 11 | 12 | 13 | 14 | 15 | 16 |
| --- | --- | --- | --- | --- | --- | --- | --- | --- | --- | --- | --- | --- | --- | --- | --- | --- | --- |
| | Single-beam | 0 | 10 | 20 | 30 | 40 | 50 | 60 | 70 | 80 | 90 | 100 | 110 | 120 | 130 | 140 | 150 |
| Working mode | Dual-beam | 0 | 10 | 20 | 30 | 40 | 50 | 60 | 70 | 35 | 30 | 25 | 20 | 15 | 10 | 5 | 0 |
| | Quadra-beam | 0 | 2.5 | 5.0 | 7.5 | 0 | 10 | 20 | 30 | 15 | 10 | 5 | 0 | 22.5 | 15 | 7.5 | 0 |

All the delay values are in ps.

### 3.1. Single-Beam Beamforming

When beamforming and transmitting only one RF signal, the input RF signal has the highest priority and can enable all AEs in the array. In this situation, the optical power distribution is shown in Figure 8a. The optical signal is inputted into the splitter array at the first stage and then split into 16 branches by controlling the optical switch array. The transmission value in every output is listed in Table 4.

**Table 4.** The transmission values in 16 outputs of single-beam mode.

| Outputs | 1 | 2 | 3 | 4 | 5 | 6 | 7 | 8 | 9 | 10 | 11 | 12 | 13 | 14 | 15 | 16 |
| --- | --- | --- | --- | --- | --- | --- | --- | --- | --- | --- | --- | --- | --- | --- | --- | --- |
| Wavelength | | | | | | | | 1550 nm | | | | | | | | |
| Transmission value $(\times 10^{-2})$ | 3.57 | 3.57 | 3.57 | 3.57 | 3.74 | 3.75 | 3.46 | 3.46 | 3.57 | 3.57 | 3.62 | 3.62 | 3.57 | 3.57 | 3.57 | 3.57 |

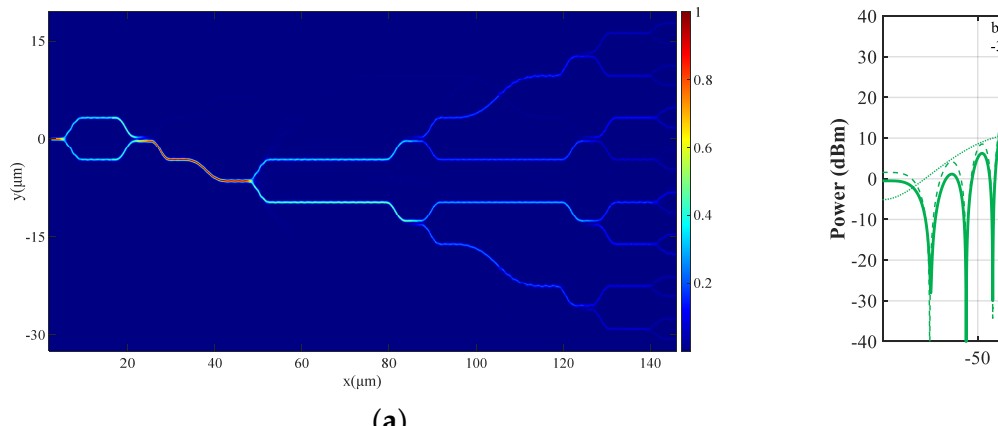
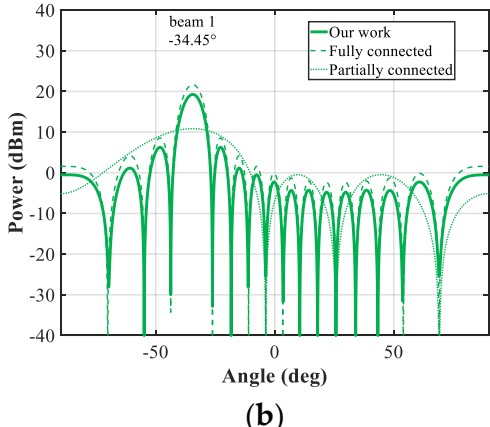

**Figure 8.** (**a**) Optical power distribution in single-beam mode; (**b**) beam pattern in single-beam mode.

It should be noted that when the system works in the single-beam mode, the transmission value in each output is $6.25 \times 10^{-2}$ under the ideal situation, which sets the corresponding AE excited power as 1 mW. Therefore, in reality, the excited power of AE is calculated by the ratio between the transmission value in Table 4 and the transmission value in the ideal situation. Additionally, the eight-phase compensation waveguides are set as [0 $\pi/2$ $\pi/2$ $\pi$ $\pi$ $\pi/2$ $\pi/2$ 0]. According to the transmission value in Table 4 and the delay value in the first row of Table 3, the beam pattern is plotted in Figure 8b. In Figure 8b, the performances of the fully connected and partially connected architecture to form the signal beam with the same number in the same direction are also plotted. The beam pattern in the dashed line represents the fully connected architecture, which is plotted according to the transmission values of $4.74 \times 10^{-2}$ in sixteen outputs. Meanwhile, the beam pattern in the dotted line represents the partially connected architecture, which is plotted according to the transmission values of $2.18 \times 10^{-1}$ in four outputs. The transmission values in the fully connected and partially connected architecture are constant because the connections between the RF signal and AEs in the two architectures are fixed.

*3.2. Dual-Beam Beamforming*

The second numerical simulation relates to a scenario where two signals are beamformed and transmitted simultaneously. In this case, the entire antenna array is divided into two subarrays. Each subarray has 8 AEs and is used to transmit one RF signal. The optical power distribution is shown in Figure 9a. Two optical signals inject the splitter array at second-stage inputs, and each optical signal is split into eight branches to enable the corresponding AEs to transmit. The transmission value in every output is listed in Table 5.

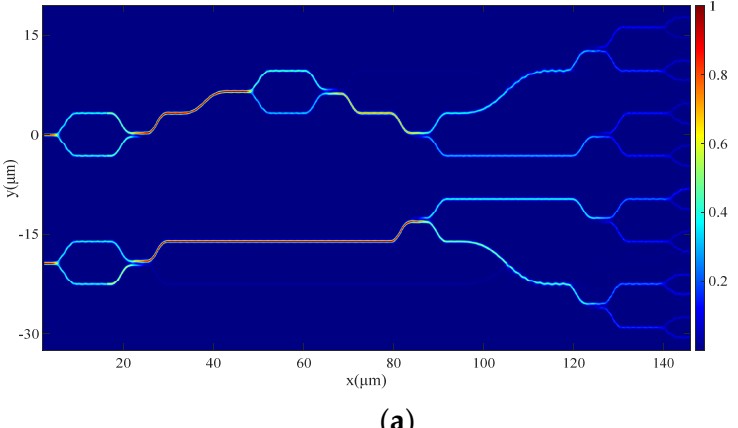
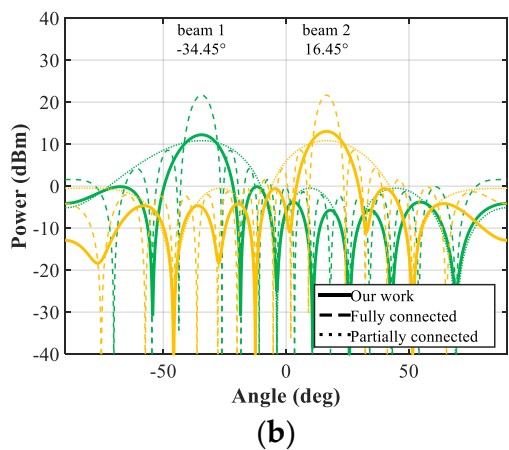

**Figure 9.** (**a**) Optical power distribution in dual-beam mode; (**b**) beam pattern in dual-beam mode.

**Table 5.** The transmission values in 16 outputs of dual-beam mode.

| Outputs | 1 | 2 | 3 | 4 | 5 | 6 | 7 | 8 | 9 | 10 | 11 | 12 | 13 | 14 | 15 | 16 |
|---|---|---|---|---|---|---|---|---|---|---|---|---|---|---|---|---|
| Wavelength | 1550 nm | | | | | | | | 1551 nm | | | | | | | |
| Transmission value ($\times 10^{-2}$) | 6.46 | 6.46 | 6.46 | 6.46 | 6.07 | 6.07 | 6.57 | 6.57 | 8.01 | 8.01 | 7.42 | 7.42 | 7.66 | 7.66 | 7.66 | 7.66 |

Similar to the situation of single-beam mode, the ideal transmission value in the dual-beam mode is 0.125. The phase compensation waveguides are set as [$\pi/2$ $\pi$ 0 $\pi/2$ $\pi/2$ 0 $\pi$ $\pi/2$]. The beam pattern is shown in Figure 9b according to the optical transmission value at each output port in Table 5 and the delay value in the second row of Table 3. The beam patterns of the fully connected and partially connected architecture in the same communication scenario are also plotted in Figure 9b.

### 3.3. Quadra-Beam Beamforming

The array is divided into four subarrays, each with four AEs when four RF signals are input. The optical power distribution is shown in Figure 10a. Four optical signals are input into the splitter array at the third stage, and each is split into four branches. The transmission value in every output is listed in Table 6.

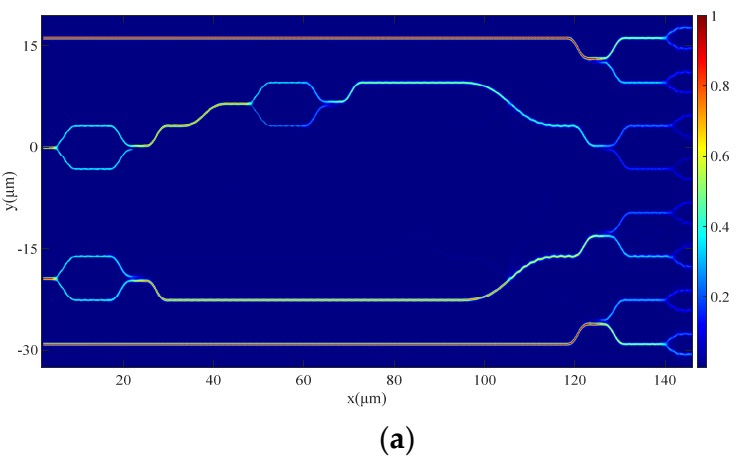

(**a**)

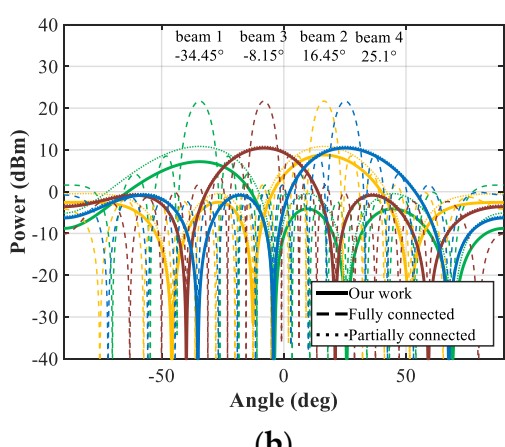

(**b**)

**Figure 10.** (**a**) Optical field distribution in quadra-beam mode; (**b**) beam pattern in quadra-beam mode.

**Table 6.** The transmission values in 16 outputs of quadra-beam mode.

| Outputs | 1 | 2 | 3 | 4 | 5 | 6 | 7 | 8 | 9 | 10 | 11 | 12 | 13 | 14 | 15 | 16 |
|---|---|---|---|---|---|---|---|---|---|---|---|---|---|---|---|---|
| Wavelength | 1552 nm | | | | 1550 nm | | | | 1551 nm | | | | 1553 nm | | | |
| Transmission value ($\times 10^{-1}$) | 2.08 | 2.08 | 2.08 | 2.08 | 1.42 | 1.42 | 1.44 | 1.44 | 1.74 | 1.74 | 1.69 | 1.69 | 2.08 | 2.08 | 2.08 | 2.08 |

Similar to the single-beam and dual-beam modes, the ideal transmission value in the quadra-beam mode is 0.25. The phase compensation waveguides are set as [$\pi/2$ 0 $\pi/2$ 0 0 $\pi/2$ 0 $\pi/2$]. The beam pattern is plotted in Figure 10b according to the transmission value in Table 6 and delay values in the third row of Table 3. The beam patterns of the fully connected and partially connected architecture in the same communication scenario are also plotted in Figure 10b.

## 4. Discussion

The simulations validate the feasibility of SDN, which offers the reconfigurable connection between the RF signals and the AEs and enables the system to reconfigure the array

subarray to transmit according to the number of input RF signals. Some discussions in detail are expanded based on the above simulations.

### 4.1. Transmission Analysis

In the photonic integrated circuit, component imperfections will affect system performance. The most significant impact is caused by the crosstalk of $1 \times 2$ optical switch elements, which creates several interferometers within the optical path. For example, in the optical path of signal 1, the interference is generated at the $2 \times 2$ coupler with two inputs connected to signal 1. The same interference occurs in the optical path of signal 2.

Set the crosstalk of the $1 \times 2$ optical switch to the typical value of 40 dB and simulate the 16 outputs transmission curves for three working modes. The simulation results are plotted in Figure 11. As seen in Figure 11, the transmission curves display slight wavelength dependence between signal 1 or 2 and certain outputs but do not impose limitations on the system broadband operation. However, during deployments, it is necessary to monitor the output of the optical switch to ensure its functionality and eliminate any optical interference. In addition, power compensation is needed because of the high insertion loss of optical components.

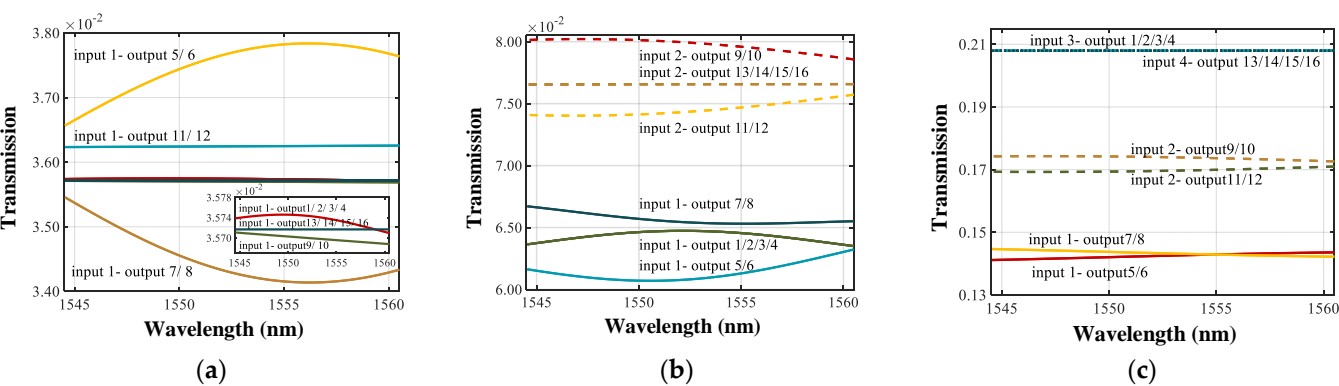

**Figure 11.** Transmission curves for three working modes: (**a**) single-beam mode, (**b**) dual-beam mode, and (**c**) quadra-beam mode.

### 4.2. Reconfiguration Time

The time required for reconfiguration in SDN is an important parameter because the main deployment situations of the optical multi-beamforming based on SDN are future wireless communication or radar systems. The reconfiguration of SDN is realized by reconfiguring the optical switch array. Therefore, the time required for reconfiguration in SDN depends on the time needed for switching a single $1 \times 2$ optical switch. The optical switches, based on different principles, have different switching times. For example, the switching time required in the thermo-optical switch is at the micro-second level [30]. As for the electronic–optical switch, the switching time is at the nano-second level [31].

In the $4 \times 16$ SDN, shown in Figure 6, three $1 \times 2$ optical switches are included. Therefore, when reconfiguring the working mode in SDN, a minimum of one $1 \times 2$ optical switch or a maximum of three $1 \times 2$ optical switches must be enabled. In the ideal situation, the time required for reconfiguration in SDN is equal to the time required for switching one single $1 \times 2$ optical switch. The $1 \times 2$ optical switch in the proposed SDN is realized in the thermo-optic effect. Many factors influence the switching time required for a thermo-optic switch, such as the material of the thermocouple, the parameter in the size of the thermocouple, and the distance between the thermocouple and the core of the waveguide. To simulate the switching time required for a single thermo-optical switch in the proposed SDN, the material of the thermocouple is set to nichrome, the size of the thermocouple is set as 2 μm in width and 0.12 μm in height, and the distance between the thermocouple and waveguide core is set as 2 μm. The simulation result is shown in Figure 12. As can be seen in Figure 12, the rise time, $\tau_{\text{rise}} = 14$ μs, is equal to the fall time, $\tau_{\text{fall}} = 14$ μs. Thus, in

the ideal conditions, the reconfiguration time of the proposed SDN is 14 µs. This value is related to the specific parameter setup of the $1 \times 2$ optical switch. In addition, in the actual deployment, the time required for reconfiguration in SDN equals the longest switching time in three $1 \times 2$ optical switches.

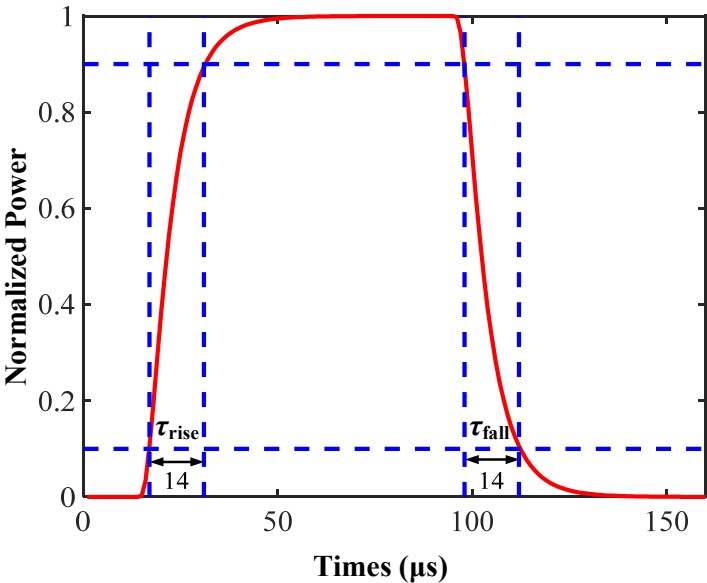

**Figure 12.** The simulation result of the $1 \times 2$ thermo-optic switch.

### 4.3. Performance Comparison

The comparisons between the proposed and the two architectures in Figure 1 are given.

The capability of beamforming is the key parameter in an optical beamforming system. To better compare the beamforming capability, three different beam pattern curves in three types of connected architecture are plotted in Figure 8b, Figure 9b, and Figure 10b, respectively. In the three beam pattern plots, the same line color indicates the signal beam pointing towards the same direction, and the different styles of the line indicate the signal beam formed in different connected architectures. The solid line indicates the beamforming capacity of the proposed reconfigurable connected architecture. The dashed line indicates the beamforming capacity of the fully connected architecture. The dotted line indicates the beamforming capacity of the partially connected architecture. Some parameters are used to evaluate the beamforming capability, such as main lobe power, full width at half maxima (FWHM), and side lobe depression.

The comparison results are listed in Tables 7–10. Some conclusions can be drawn from the four tables.

**Table 7.** Beam pattern parameter comparison ($\theta_1 = -34.45$ deg).

| Parameter | Fully Connected | Partially Connected | Our Work | | |
|---|---|---|---|---|---|
| | | | Working Mode | | |
| | | | Single-Beam | Dual-Beam | Quadra-Beam |
| Main lobe power (dBm) | 21.68 | 10.84 | 19.26 | 12.24 | 7.18 |
| FWHM (deg) | 7.70 | 32.65 | 7.70 | 15.45 | 32.65 |
| Sidelobe depression (dB) | 13.15 | 11.30 | 13.01 | 12.36 | 11.30 |

**Table 8.** Beam pattern parameter comparison ($\theta_2$ = 16.45 deg).

| Parameter | Fully Connected | Partially Connected | Our Work | |
|---|---|---|---|---|
| | | | Working Mode | |
| | | | Dual-Beam | Quadra-Beam |
| Main lobe power (dBm) | 21.68 | 10.84 | 13.05 | 8.78 |
| FWHM (deg) | 6.60 | 27.50 | 13.60 | 27.50 |
| Sidelobe depression (dB) | 13.16 | 11.30 | 12.81 | 11.30 |

**Table 9.** Beam pattern parameter comparison ($\theta_3$ = −8.15 deg).

| Parameter | Fully Connected | Partially Connected | Our Work |
|---|---|---|---|
| | | | Working Mode |
| | | | Quadra-Beam |
| Main lobe power (dBm) | 21.68 | 10.84 | 10.44 |
| FWHM (deg) | 6.40 | 26.60 | 26.60 |
| Sidelobe depression (dB) | 13.15 | 10.38 | 10.38 |

**Table 10.** Beam pattern parameter comparison ($\theta_4$ = 25.10 deg).

| Parameter | Fully Connected | Partially Connected | Our Work |
|---|---|---|---|
| | | | Working Mode |
| | | | Quadra-Beam |
| Main lobe power (dBm) | 21.68 | 10.84 | 10.44 |
| FWHM (deg) | 7.00 | 29.30 | 29.30 |
| Sidelobe depression (dB) | 13.16 | 11.30 | 11.30 |

Firstly, the proposed reconfigurable connected architecture in single-beam working mode has the same beam width and a similar sidelobe depression with fully connected architecture by comparing the data in Table 7. However, the main lobe power of the proposed architecture is lower by 2.42 dB than the fully connected architecture. The reason is the high loss of the optical component in the SDN optical path.

Secondly, the proposed reconfigurable connected architecture in quadra-beam working mode has the same beam width and sidelobe depression as the partially connected architecture by comparing the data between four tables. Whereas, because of the high loss of optical components in the SDN optical path, the main lobe powers of the proposed reconfigurable connected are lower by 3.66, 2.06, 0.40, and 0.40 dB than the partially connected architecture, respectively. Therefore, based on the above two points, optical amplifiers are needed in the SDN optical path to compensate for the optical power and enhance the beamforming capability of the reconfigurable optical multi-beamforming based on SDN.

Lastly, by comparing the parameters of the proposed reconfigurable connected architecture in different working modes between four tables, the proposed reconfigurable connected architecture in single-beam mode has the highest main lobe power, the smallest beam width, and the highest sidelobe depression than the other working mode. Moreover, the proposed reconfigurable connected architecture in quadra-beam mode has the lowest main lobe power, the biggest beam width, and the lowest sidelobe depression. Therefore, the ability to focus the power in a signal beam is diminishing with the increasing number of input RF signals in the proposed reconfigurable architecture, which also means the reconfigurable multi-beamforming based on SDN has the variable beamforming gain, can reconfigure the capability of beamforming and adapting to the various RF signal transmission scenarios.

In addition to comparing the beamforming capability, the number of optical components in the connected architecture was also evaluated. These components include splitter elements and switch elements. The results are presented in Table 11 below.

**Table 11.** Number of optical components comparison.

| Components Type | Fully Connected | Partially Connected | Our Work |
|---|---|---|---|
| Optical splitter | $N_{RF}(N_{AE}-1)$ | $N_{AE}-N_{RF}$ | $N_{AE}-1$ |
| Optical switch | 0 | 0 | $N_{RF}-1$ |

To instinctively comprise the number of optical components included in three architectures, the parameter sweeps with $N_{AE}$ = 16, 32, 64, 128, and $N_{RF}$ = 4, 8, 16 are performed, and the bar graph is shown in Figure 13.

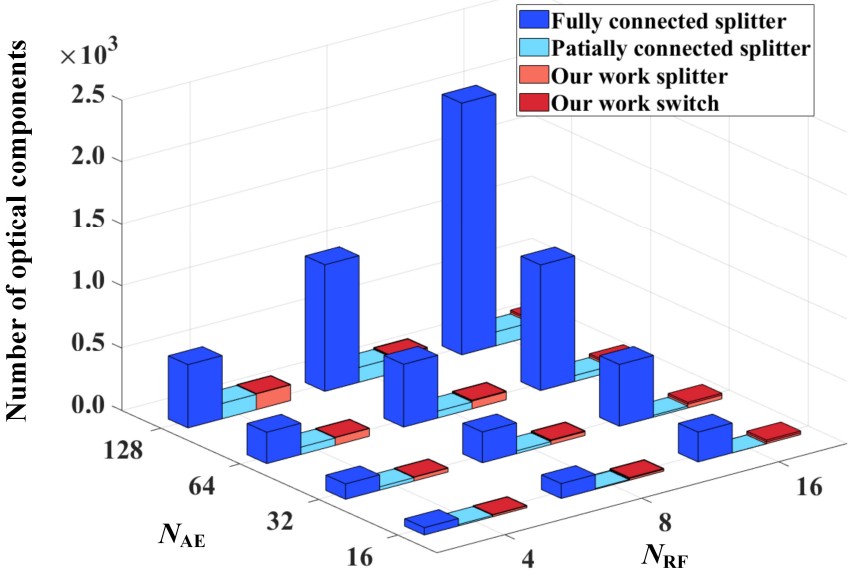

**Figure 13.** The number of optical components comparison bar graph.

As can be seen in Figure 13, the fully connected architecture has the greatest number of optical splitters compared to the others, which means it has the highest hardware complexity. The gaps in hardware complexity between the fully connected architecture and the other two architectures are increased with the number of input signals $N_{RF}$ and the number of antenna elements $N_{AE}$ increasing. Furthermore, the partially connected architecture has an approximate number of optical components similar to the proposed architecture.

In summary, compared to the fully connected architectures, the reconfigurable multi-beamforming system based on SDN maintains the highest beamforming gains while significantly reducing the hardware complexity. Compared to the partially connected architecture, the proposed architecture exhibits equivalent hardware complexity but can achieve greater beamforming gains. Thus, the proposed reconfigurable optical multi-beamforming system based on SDN offers more degrees of freedom and adaptability in a multitude of communication scenarios for the future.

*4.4. Beam Steering Capability*

The beam steering pattern simulations are performed to prove the integration of SDN will not influence the beamforming and steering capability of the OTTD array. In the parameter setup of Section 3, the OTTD devices employed in our proposed system are the same as the devices in the architecture of Figure 1, which can form eleven beams in different directions to cover the sector in ninety degrees. Figure 14 is plotted to demonstrate the beam steering capability. Figure 14a describes the beam steering of RF signal 1 with eleven beam directions when the proposed system works in single-beam mode. Figure 14b illustrates the beam steering of RF signal 2 with eleven beam directions when the system

works in dual-beam mode. Figure 14c describes the beam steering of RF signal 3 (which equals RF signal 4) with eleven beam directions when the system works in quadra-beam mode. The beam parameters, such as main lobe power, FWHM, and side lobe depression under different situations, are listed in Table 12, Table 13, and Table 14, respectively.

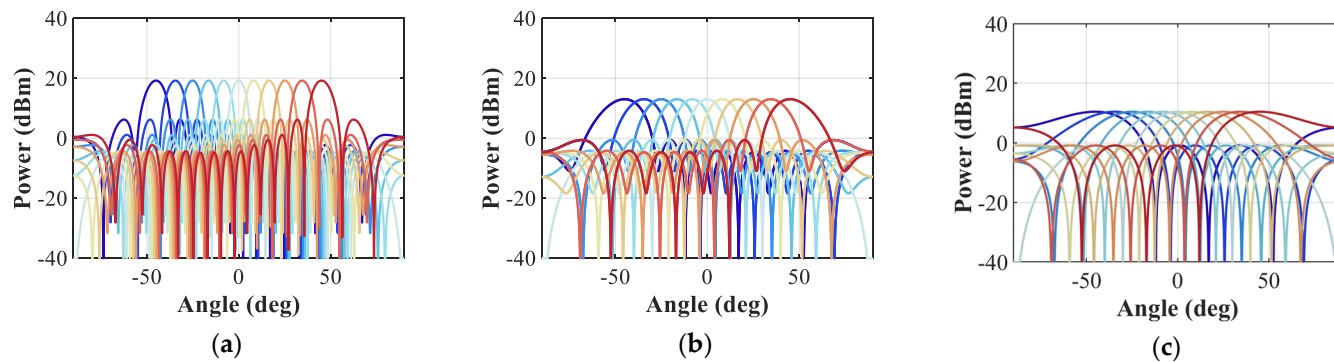

**Figure 14.** Beam steering pattern of different beam directions: (**a**) single-beam mode, (**b**) dual-beam mode, and (**c**) quadra-beam mode.

**Table 12.** Beam parameter of the RF signal 1 in single-beam mode.

| Beam Direction (deg) | −45.00 | −34.45 | −25.10 | −16.45 | −8.15 | 0.00 | 8.15 | 16.45 | 25.10 | 34.45 | 45.00 |
|---|---|---|---|---|---|---|---|---|---|---|---|
| Main lobe power (dBm) | 19.26 | 19.26 | 19.26 | 19.26 | 19.26 | 19.26 | 19.26 | 19.26 | 19.26 | 19.26 | 19.26 |
| FWHM (deg) | 9.05 | 7.07 | 7.00 | 6.60 | 6.40 | 6.30 | 6.40 | 6.60 | 7.00 | 7.07 | 9.05 |
| Sidelobe depression (dB) | 13.01 | 13.01 | 13.01 | 13.01 | 13.01 | 13.01 | 13.01 | 13.01 | 13.01 | 13.01 | 13.01 |

**Table 13.** Beam parameter of the RF signal 2 in dual-beam mode.

| Beam Direction (deg) | −45.00 | −34.45 | −25.10 | −16.45 | −8.15 | 0.00 | 8.15 | 16.45 | 25.10 | 34.45 | 45.00 |
|---|---|---|---|---|---|---|---|---|---|---|---|
| Main lobe power (dBm) | 13.05 | 13.05 | 13.05 | 13.05 | 13.05 | 13.05 | 13.05 | 13.05 | 13.05 | 13.05 | 13.05 |
| FWHM (deg) | 18.75 | 15.95 | 14.45 | 13.60 | 13.20 | 13.10 | 13.20 | 13.60 | 14.45 | 15.95 | 18.75 |
| Sidelobe depression (dB) | 12.81 | 12.81 | 12.81 | 12.81 | 12.81 | 12.81 | 12.81 | 12.81 | 12.81 | 12.81 | 12.81 |

**Table 14.** Beam parameter of the RF signal 3 in quadra-beam mode.

| Beam Direction (deg) | −45.00 | −34.45 | −25.10 | −16.45 | −8.15 | 0.00 | 8.15 | 16.45 | 25.10 | 34.45 | 45.00 |
|---|---|---|---|---|---|---|---|---|---|---|---|
| Main lobe power (dBm) | 10.44 | 10.44 | 10.44 | 10.44 | 10.44 | 10.44 | 10.44 | 10.44 | 10.44 | 10.44 | 10.44 |
| FWHM (deg) | 40.50 | 32.65 | 29.35 | 27.45 | 26.55 | 26.30 | 26.55 | 27.45 | 29.35 | 32.65 | 40.50 |
| Sidelobe depression (dB) | 11.30 | 11.30 | 11.30 | 11.30 | 11.30 | 11.30 | 11.30 | 11.30 | 11.30 | 11.30 | 11.30 |

The powers of the main lobe and side lobe are constant over the direction of the beam by comparing the beam parameters in different directions. Meanwhile, the beam width will expand with the beam direction away from the normal, which means the radiative power focusing ability of the antenna array decreases as the beam pointing angle increases. As shown in Figure 14, the different signals all formed beams pointing in eleven different directions under different SDN working modes. Therefore, the multi-beamforming architecture based on SDN can provide continuous beam steering.

## 5. Conclusions

A reconfigurable optical multi-beamforming based on the switching and distributing network has been proposed and simulated. The reconfigurable connections between RF signals and antenna elements are established by integrating the optical switch arrays and splitter array into the multi-beamforming architecture. The proposed architecture allows reconfiguring the antenna subarray to beamform and transmit the RF signals depending

on the number of input RF signals. An instance of the proposed architecture, supporting 4 RF signals beamforming and transmitting through 16 antenna elements, is demonstrated. A series of numerical simulations are conducted to evaluate its performance and validate its feasibility. The impact of optical component imperfections and the comparison with two typical fixed-connected architectures are discussed. The results show that the proposal requires compensation for amplitude–phase inconsistency and monitoring the working state of the optical switch. The advantages of the proposed architecture, which possesses a variable beamforming gain and lower hardware complexity, have been verified by comparing it to two fixed-connected architectures. The proposed approach satisfies the requirements of future communication and radar applications, such as antenna selection technology and multi-functional radar.

**Author Contributions:** Conceptualization, Q.Q.; methodology, Y.L. and Q.Q.; software, Y.L. and D.J.; validation, Y.L. and X.L..; formal analysis, Y.L. and D.J.; investigation, Y.L.; resources, Y.L. and Q.Q.; data curation, Y.L., D.J., Y.C., X.L. and Q.Q.; writing—original draft preparation, Y.L.; writing—review and editing, Y.L., D.J., Y.C., X.L. and Q.Q.; visualization, Y.L.; supervision, Q.Q.; project administration, Q.Q.; funding acquisition, Q.Q. All authors have read and agreed to the published version of the manuscript.

**Funding:** This work was supported by the National Natural Science Foundation of China (62201120, 61971110), the Research Foundation (Y030212059003044), and the Natural Science Foundation of Sichuan Province (2023NSFSC1379, 2023NSFSC0448).

**Institutional Review Board Statement:** Not applicable.

**Informed Consent Statement:** Not applicable.

**Data Availability Statement:** Further data are available from the corresponding author upon reasonable request.

**Acknowledgments:** The authors would like to acknowledge the support of the National Natural Science Foundation of China, the Research Foundation, and the Natural Science Foundation of Sichuan Province.

**Conflicts of Interest:** The authors declare no conflicts of interest.

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
