# Peer review of "Reconfigurable Microwave Multi-Beamforming Based on Optical Switching and Distributing Network"

_photonics, doi:10.3390/photonics11010065_

Round 1
Reviewer 1 Report (Previous Reviewer 1)
Comments and Suggestions for Authors
This manuscript has been fully revised. The questions have been answered clearly. In the revision, the authors provided a convincing explanation of the reconstruction time and compared the beamforming gain in detail.
One minor issue: it appears to me that this is still be microwave beamforming using photonic techniques. Therefore, the current title is not proper. A better title could be "Reconfigurable Microwave Multi-Beamforming Based on Switching and Distributing Network."
Overall, the quality of the revised manuscript has been improved and I believe it is suitable for publication in Photonics after revising the title.
See comments above about the title.
Author Response
Please see the attachment.

Reviewer 2 Report (Previous Reviewer 2)
Comments and Suggestions for Authors
Comments to the authors:
1. When discussing a figure, define the parameters that are appeared within the figure. For example, in Fig. 1 define N_RF and N_AE in caption or within the text.
2. Line 80, page 2: There is a type “conventional fixe architectures are”.
3. Define V_\pi in (1) and (5). Add a reference for (1) and (5).
4. Page 5, first paragraph, please use “inject the signal” instead of “access the signal”.
5. In Table 1 and in the text, it is not clear whether signals at different priority ranks are connected to all antennas, or a higher rank means connection to a smaller number of antennas.
6. Continuous beam steering pattern properties must be demonstrated. For example, normalized peak pattern and SLL versus steering angle when the beam is continuously steered. If the beam directions are fixed, the presented method has no merit as the architectures in Figure 1 could provide continuous beam steering.
7. Since the paper does not have any measurement, the details about the numerical simulation must be included in the paper instead of just referring to another publication. Response of individual components must be added.
8. Figure 12 is too small.
Round 2
Reviewer 2 Report (Previous Reviewer 2)
Comments and Suggestions for Authors
The authors addressed all my comments.
This manuscript is a resubmission of an earlier submission. The following is a list of the peer review reports and author responses from that submission.
Round 1
Reviewer 1 Report
Comments and Suggestions for Authors
This manuscript presents an innovative reconfigurable design and simulation for the OTTD structure used for antenna arrays, which finds a good balance between hardware complexity and beamforming gain. As detailed in the manuscript, the transmitter system with reconfigurable subarrays can adapt to the trend of future wireless communication. This study seems interesting and meaningful, yet there are certain aspects of the content that might be confusing:
1. The authors refer to the control complexity denoted as 'G' and provide a calculation formula in (1). However, the manuscript lacks an explanation or proper citations for this particular formula. Incorporating relevant explanations and citing any sources that contribute to the development of this formula would enhance the clarity and credibility of the manuscript. In addition, comparing the G values of the other two OTTD structures with the results can highlight the advantage of this work.
2.The simulation section of the manuscript does not appear to address the consideration of the time required for reconfiguration. Please provide more detailed information, such as the time required to switch between 1x16, 2x8, and 4x4 modes.
3.Fig.7(b): There seems to be no relevant explanation? Do the thin yellow lines represent wires or control signals? As a layout diagram, it lacks relevant information such as the types of devices and waveguides.
4.Table.1.2.3: The thicknesses of the lines in the middle of the tables are uneven.
5.From line 269, it seems that the selection of optical carriers is based on different wavelengths of light. Does this mean that the mode selection in the manuscript specifically refers to selecting the wavelength of light? If so, please state that since there are too many modes for light.
6.In the conclusion, the authors state that this work has higher beamforming gain than the subarray structure, but there seems to be no comparison in the manuscript?
I would like to recommend publication after the authors have considered the above questions.
Reviewer 2 Report
Comments and Suggestions for Authors
1. The paper is poorly written. The proposed method is not clearly described and fully evaluated. The paper is not easy to follow and understand because of poor English writing.
2. Description of Figure 4 and the method is vague. Please add more comments.
3. How do you compensate for phase shifts introduced by couplers?
4. What is the performance over a wide bandwidth? The purpose of the method is operation over a wide bandwidth, but it is completely neglected in numerical results.
5. The numerical results are not acceptable. It is not described how the results are generated. At least, simulation with realized components (coupler, switches) are required to evaluate the performance and feasibility of the method.
Comments on the Quality of English LanguageThe paper is not easy to follow and understand because of poor English writing.